# *In Silico* Analysis of Glutamate Receptors in *Capsicum chinense*: Structure, Evolution, and Molecular Interactions

**DOI:** 10.3390/plants13060812

**Published:** 2024-03-12

**Authors:** Fabiola León-García, Federico García-Laynes, Georgina Estrada-Tapia, Miriam Monforte-González, Manuel Martínez-Estevez, Ileana Echevarría-Machado

**Affiliations:** Unidad de Biología Integrativa, Centro de Investigación Científica de Yucatán, Calle 43, #130, x 32 and 34, Mérida 97205, Yucatán, Mexico; fabiolaleong@outlook.com (F.L.-G.); mmg@cicy.mx (M.M.-G.); luismanh@cicy.mx (M.M.-E.)

**Keywords:** habanero pepper, glutamate receptor, structure, phylogeny, ligand, molecular docking

## Abstract

Plant glutamate receptors (GLRs) are integral membrane proteins that function as non-selective cation channels, involved in the regulation of developmental events crucial in plants. Knowledge of these proteins is restricted to a few species and their true agonists are still unknown in plants. Using tomato SlGLRs, a search was performed in the pepper database to identify GLR sequences in habanero pepper (*Capsicum chinense* Jacq.). Structural, phylogenetic, and orthology analysis of the CcGLRs, as well as molecular docking and protein interaction networks, were conducted. Seventeen CcGLRs were identified, which contained the characteristic domains of GLR. The variation of conserved residues in the M2 transmembrane domain between members suggests a difference in ion selectivity and/or conduction. Also, new conserved motifs in the ligand-binding regions are reported. Duplication events seem to drive the expansion of the species, and these were located in the evolution by using orthologs. Molecular docking analysis allowed us to identify differences in the agonist binding pocket between CcGLRs, which suggest the existence of different affinities for amino acids. The possible interaction of some CcGLRs with proteins leads to suggesting specific functions for them within the plant. These results offer important functional clues for CcGLR, probably extrapolated to other Solanaceae.

## 1. Introduction

Ionotropic glutamate receptors (iGluRs) are integral membrane proteins that function as ligand-gated nonselective cation channels (NSCC). Initially, these proteins were discovered in the central nervous system (CNS) of mammals, where the interaction with their agonists induced the opening of the channel and the passage of cations such as Ca^2+^, Na^+^, and K^+^ [1]. However, proteins with high similarity to iGluRs have been discovered in plants for more than two decades and, therefore, were named GLR (glutamate-like receptor) [2].

Since the discovery of 20 GLR sequences in the *Arabidopsis thaliana* genome [2,3], these have been identified in different plant species, demonstrating that they are a multigene family. At the same time, 13 GLRs have been reported in *Solanum lycopersicum* (SlGLR) [4], 24 GLRs in *Oryza sativa* (OsGLR) [5,6], 29 GLRs in *Medicago truncatula* (MtGLR) [7], 143 GLRs in the woody plants *Pyrus bretschneideri* (34 PbrGLR), *Fragaria vesca* (36 FveGLR), 34 GLRs *Prunus mume* (34 PmuGLR) and *Prunus persica* (40 PpeGLR) [8], 34 GLRs in *Saccharum spontaneum* (SsGLR) [9], 36 GLRs from *Gossypium hirsutum* (GhGLRs) [10] and 16 GLRs from *Zea mays* (ZmGLR) [9].

At a structural level, these proteins have an N-terminal domain (NTD) oriented towards the extracellular side, a ligand-binding domain (LBD) that contains two subunits, S1 and S2, and four transmembrane domains (TMD, M1-M4), where M2 does not completely cross the membrane and a C-terminal domain (CTD) located on the cytosolic side [2]. This structure is conserved from the iGluRs, especially in the membrane regions; the TMDs of AtGLR present 64% identity with those iGluR of the NMDA (N-methyl-D-aspartic) type from animals, while the LBD of AtGLR only has 16% identity with those from animals [2,11,12].

Phylogenetic analysis suggests the classification of *A. thaliana* genes into three different clades (clades I, II, and III) that diverge from NMDA and non-NMDA iGluRs [3]. Tomato consists of three clades (clades I, II, and III) and shares clades II and III with *A. thaliana*; the exception is clade I, which is located on a different branch than clade I of the Brassicacea [4]. For their part, the rice GLR sequences are classified into four families (clade I, II, III, and IV); while the first three clades seem to have a common ancestor with *A. thaliana*, clade IV is apart from all the others suggesting a possible gene duplication [5]. In *M. truncatula*, GLR sequences were classified into four groups (clade II, III, IV, and V); two of these clades (clade IV and V) emerged separately from clades I, II, and II of *A. thaliana* and are specific to *M. truncatula*, while clades II and III were grouped together in the same families II and III of Brassicacea. Clade I was specific to *A. thaliana* [7]. Woody species have also been analyzed by phylogeny [8] and, in these, the GLRs were also classified into four clades (I, II, III, and IV), the first three clades were shared with *A. thaliana* and the fourth clade was specific to woody species [8].

It is suggested that for the channel to be functional in the receptor, it must be made up of at least four subunits, forming homo- or heterotetramers, although the structure of the functional receptor complex has not yet been elucidated [13,14,15]. However, the interaction of several subunits belonging to different clades has been demonstrated experimentally [6,13,15]. Other ways in which the activity of these proteins is regulated are by phosphorylation mediated by their binding to 14-3-3 proteins [16,17], N-glycosylation [11,16,18,19], and compartmentalization [15,20,21].

GLRs participate in numerous processes in plants, such as the regulation of C and N metabolism [22,23], and water balance [22], responses to light [2,24], the control of Ca^2+^ signaling [25], the regulation of abscisic acid (ABA) biosynthesis [22], the coordination of mitotic activity during root development [6,26], pollen tube growth [27], response to aluminum toxicity [28], regulation of L-Met-induced membrane depolarization in *A. thaliana* leaves [29], defense signals [30], regulation of gravitropism in the roots [31], and the development of the RLs [15].

Plant GLRs are more versatile than mammalian GLRs in terms of their binding capacity between different subunits and their ability to interact with different agonists; this could be of great importance in the variability of plants to respond to environmental changes. However, there is very little information related to true GLR agonists. As in the case of mammalian iGluRs, L-Glu is considered the main agonist of GLRs present in plants [32]. However, unlike those of mammals, other amino acids (AAs), such as D-Ser [27], L-Met, L-Trp, L-Phe, L-Leu, L-Tyr, L-Asp, and L-Thr [29], and the tripeptide glutathione [33,34] have been proposed as possible agonists. On the other hand, through in silico molecular docking, possible agonists of GLRs 2.9 and 1.1 of *A. thaliana* have been demonstrated; Gly docked to the ligand-binding pocket in the AtGLR2.9 receptor; this was not the case with L-Glu. On the other hand, Gly and L-Glu were coupled to AtGLR1.1 [35].

More recently, authors [32] biochemically reconstituted the LBD of AtGLR3.3 and analyzed its amino acid binding selectivity profile, finding that two sulfur-containing AAs (L-Cys, L-Met), as well as L-Glu and Gly, bound to the ligand-binding pocket in AtGLR2.9, and that L-Cys and L-Met had the best dissociation constant; that is, they bound with greater affinity to the receptor in relation to the other AAs. Similar to this case, Gly and L-Met are able to bind to the binding pocket of AtGLR3.2 [36] and L-Glu and glutathione are AAs that can bind to the LBD of AtGLR3.4 [37].

The Solanaceae are a family of plants that comprises approximately 98 genera and about 2700 species. Among the most important food species belonging to this family are potato (*Solanum tuberosum*), eggplant (*Solanum melongena*), tomato (*S. lycopersicum*), and chili peppers (*Capsicum*) [38]. The *Capsicum* genus comprises approximately 35 species that exhibit high genetic and phenotypic diversity, considering five of them as domesticated and economically important species: *C. annuum*, *C. baccatum* var. pendulum, *C. chinense* Jacq. (Habanero pepper), *C. frutescens* L., and *C. pubescens* Ruiz and Pav. [39,40]. The wide use of chili peppers, from culinary to industrial and health uses, has led to their worldwide cultivation, and although their total global production has been increasing, it does not seem to be enough to satisfy the growing demand for these chili peppers [41]. Their demand is due, in part, to the fact that these species are rich in bioactive compounds that have beneficial properties, with carotenoids and capsaicinoids being two of the majority compounds [40].

Habanero pepper is one of the most widely diverse domesticated species of this genus, considered one of the hottest chili peppers in the world, and highly demanded in the market, due to its aroma and flavors. Likewise, this species has been shown to exhibit protective effects, including antioxidant and anticancer properties [42,43,44,45], suggesting its potential to reduce or prevent chronic diseases [46]. The great diversity in the shape, size, and color of its fruits indicates the broad genetic structure that this species can present, with the northern lowlands of the Amazon basin being considered the center of its diversification [47]. The environmental conditions in which habanero pepper grows significantly influence the development and metabolic composition of this crop, ultimately affecting the quality of its fruits and yields [48,49,50,51]. For example, soils that differ in nutrient content cause a differential effect, affecting production and gene expression [49,52]. However, the molecular mechanisms underlying the regulation of growth and development of this species are largely unknown.

In previous studies, we showed that the exogenous addition of AAs, possible agonists of GLRs, significantly modify the root growth of habanero pepper in a different way than in the model plant *A. thaliana* [53,54]. We suggest that GLRs could participate in this response, as has been previously demonstrated for *A. thaliana* [6,15,26]. However, GLRs have not yet been identified in habanero pepper. Tomato has been the only member of the Solanaceae in which GLRs have been studied [4,55] and, interestingly, some SlGLRs seem to be placed in evolutionarily different clades than AtGLRs, which could suggest differences in their functions between the two species. These previous results strengthen the importance of the study of these proteins in habanero pepper to provide knowledge of them in a family of plants that contains members of great agricultural importance, and whose GLRs could regulate key processes of growth and development. The objective of this work was to identify and characterize the GLR gene family of this species, as well as to identify possible agonists of the same through molecular docking analysis.

## 2. Results

### 2.1. Identification of the GLR Gene Family in the Capsicum Chinense Genome

To identify genes like habanero pepper glutamate receptors, a BLASTP search was performed using the file of habanero pepper coding sequences obtained from the pepper database (Pepper Genome platform) and downloaded into the BioEdit Sequence Alignment Editor Software version 7.2.5 (See Section 4). SlGLRs have been previously characterized structurally [4] and functionally [56]. For this reason and taking into account that tomatoes belong to the Solanaceae like habanero peppers, these sequences were used as a query. Those sequences with the highest scoring hit were considered candidates. After removing redundant sequences between different analyses, 25 candidates for habanero pepper glutamate receptor-like genes (CcGLR) were identified. Of these, six sequences were excluded because they did not meet the e-value criteria.

As a result of the analysis of the protein domains in the NCBI Conserved Domain Search Platform, the remaining 19 CcGLRs sequences presented the key functional domains, PBP1_GABAb_receptor (ligand-binding domain of GABAb receptors) and GluR_Plant (glutamate receptor domain in plants) (Appendix A). However, when the prediction of transmembrane regions was carried out using the TMPred and TMHMM bioinformatics programs, it was determined that the PHU20800.1 and PHU19527.1 sequences completely or partially lacked the four transmembrane regions, respectively. Also, the results of this analysis were inconclusive for three other sequences, for which it was not possible to predict the M4 transmembrane region (Appendix A). To clarify this result, four sequences of AtGLRs (AtGLR1.1, AtGLR2.1, AtGLR3.1, and AtGLR3.4), whose presence of the transmembrane domains of the protein have been previously established [2], were subjected for analysis to both platforms. Likewise, the results were inconclusive for two of them: AtGLR2.1 (the M4 region is absent) and AtGLR3.1 (M2 is absent with the TMPRED program) (Appendix A).

To resolve these discrepancies and determine the presence of the M1-M4 transmembrane domains, as well as the ligand-binding (LBD, S1, and S2), amino (NTD), and carboxy-terminal (CTD) domains in the CcGLRs proteins, a multiple alignment (see Materials and Methods) was carried out using as reference the previously characterized *A. thaliana* proteins, AtGLR3.2, AtGLR3.3 and AtGLR3.4 [2,6,7,57]. These results showed the presence of all the characteristic domains of GLRs in 17 of the 19 previously identified CcGLR sequences. It was identified that the sequence PHU20800.1 presented only the NTD and S1 domains, while PHU19527.1 contained the NTD, S1, M1, and M2 regions (Appendix A). These two sequences were discarded in subsequent studies, using only the 17 full-length CcGLRs sequences.

The analysis aimed at identifying a greater number of CcGLRs using the “GluR_Plant” domains of tomato GLRs did not produce additional sequences.

### 2.2. Structural Analysis and Main Characteristics of the CcGLR Family

The size of the 17 CcGLRs sequences ranged between 2703 and 3396 bp, encoding predicted proteins of between 901 and 1132 AAs residues, with predicted molecular masses between 100 and 127 KDa, respectively, and an isoelectric point (pI) that varies between 5.88 to 8.69, with an average of 7.14 (Table 1). Sixteen of the 17 CcGLRs have a predicted subcellular localization only in the plasma membrane (PM), with CcGLR2.5 being not only in the PM but also in the nucleus (Table 1).

Also, the presence of a signal peptide in the NTD was detected in 10 CcGLRs, spanning from 15 to 33 AAs (Table 1), while a single CcGLR (CcGLR3.3) contained the presence of a nuclear localization motif (NLS), DKSNRGSKRR (Appendix A). Interestingly, two CcGLRs, CcGLR2.5 and CcGLR2.7, presented a higher NTD (635 and 594 AAs, respectively) than the rest of the proteins, which had a length between 407 and 492 AAs (Appendix A). CTD length was also variable among CcGLRs; CcGLR3.1 presented the lowest value with 59 AAs residues, while CcGLR2.7 and CcGLR2.8 had the greatest lengths with 124 residues (Appendix A).

The percentage of identity between the proteins was lower in the NTD (38%) and CTD (21%) ends, while the most conserved domains were the TMD, particularly M3 with an average of 61% identity among all sequences. For the S1 and S2 domains, the identity percentages were higher between proteins belonging to the same clade, these being an average of 46 and 45%, respectively. In general, the percentage of identity, considering the complete sequences of the proteins, varied between 27% (between CcGLR1.3 and CcGLR3.1) and 84% (between CcGLR2.2 and CcGLR2.3).

Regarding the M2 domain, all CcGLRs presented a conserved Phe, which coincides with the F654 of AtGLR3.4. Similarly, 15 of the 17 CcGLRs presented a P (89%), coinciding with the P663 of AtGLR3.4, while CcGLR3.1 and CcGLR3.5 contain a Lys (K) and a Thr (T) in this position, respectively (Figure 1 and Appendix A). On the other hand, the highly conserved XXTAXLXS domain was present in the M3 region of the CcGLRs, with 100% conserved (SYTASLTS) for 10 CcGLRs, which include all members of clade III, 5 members of clade II (CcGLR2.1, CcGLR2.2, CcGLR2.4, CcGLR2.5, and CcGLR2.6) and CcGLR1.2 as the only member of clade I. The rest of the members of clade II presented the XYTASLXS motif, the first position being a T for CcGLR2.7, CcGLR2.8, and CcGLR2.9 or N for CcGLR2.3, and, in the seventh position, an S for the first three CcGLRs or a T for this last member. The rest of the members of clade I were the most variable, with respect to the previous highly conserved motif, being for CcGLR1.1, CcGLR1.3, and CcGLR1.4 the CFTAXLSS domain, where the fifth position varied to K, V or L, respectively. LBD S1 and S2 exhibit the DXXGD and SPLXXDXS motifs, respectively (Figure 1 and Appendix A).

Several phosphorylation sites were identified in the 17 CcGLRs proteins, varying in number from 9 (CcGLR4.4 and CcGLR3.1) to 26 (CcGLR2.4 and CcGLR2.5). Significantly, it was observed that CcGLR2.4 presented many predicted phosphorylation sites in the CTD, compared to CcGLR4.3 and CcGLR4.4, which presented exclusively a site unique (Appendix A).

### 2.3. Structure of CcGLRs Genes

Family IV presented almost all its members with five exons and four introns (*CcGLR4.2*, *CcGLR4.3*, and *CcGLR4.4*). Only CcGLR4.1 contained eight exons and seven introns. All members of family III of the CcGLRs presented six exons and five introns. Finally, family II was the one that presented the most heterogeneity: six sequences (*CcGLR2.1*, *CcGLR2.2*, *CcGLR2.3*, *CcGLR2.4*, *CcGLR2.8*, and *CcGLR2.9*) presented five exons and four introns, two sequences (*CcGLR2.5* and *CcGLR2.6*) with six exons and five introns, and a sequence with nine exons and eight introns (*CcGLR2.7*) (Table 1).

The seventeen CcGLRs sequences were identified in six of the twelve habanero pepper chromosomes: five sequences on chromosome 6 (*CcGLR2.1*, *CcGLR2.2*, *CcGLR2.3*, *CcGLR2.4,* and *CcGLR2.5*), four sequences on chromosome 4 (*CcGLR1.1*, *CcGLR1.3*, *CcGLR1.4,* and *CcGLR3.3*), three sequences on chromosome 3 (*CcGLR2.7*, *CcGLR2.8,* and *CcGLR2.9*), two other sequences were located at 2 (*CcGLR4.2* and *CcGLR3.5*) and 7 (*CcGLR3.1* and *CcGLR3.2*) chromosomes, and, finally, one sequence, *CcGLR2.6*, was located on chromosome 8 (Figure 2).

The results showed ten duplication events of the *CcGLRs*, of which eight were tandem duplications and two were segmental (Table 2). The greatest number of duplication events was observed within family II, where seven of the ten events were identified. In the case of family IV, only one duplication event occurred (CcGLR4.4 and CcGLR4.3), with the last two occurring within family III (CcGLR3.1 and CcGLR3.2 and CcGLR3.3 and CcGLR3.5) (Table 2).

By comparing the results of the duplication events with the taxonomic tree (Figure 3), we can deduce the origin of the *CcGLRs*. The two most ancestral duplication events occurred in family III. The first occurs in the paralogs *CcGLR3.3* and *CcGLR3.5*, whose orthologs are found in *Marchantia polymorpha* and *Physcomitrium patens*, respectively. However, it cannot be deduced which of them was the one duplicated because in the timeline the two are identified as the oldest species. The second duplication event is represented by *CcGLR3.1* and *CcGLR3.2*, of which we can identify an ortholog of CcGLR3.2 in *Amborella trichopoda* as the most evolutionary distant and as the precursor of CcGLR3.1 through a duplication event (Figure 3, Table 2).

In the case of family II, CcGLR2.2 is the first sequence that was identified as an ortholog of a monocot sequence (*Z. mays*) and is responsible for giving rise to *CcGLR2.3* and *CcGLR2.1* through tandem duplications. The duplication events mentioned were the most ancient.

Later duplication events of family II were more recent and possibly occurred in the Solanaceae. *CcGLR2.3*, through tandem and segmental duplication, gives rise to *CcGLR2.4* and *CcGLR2.7*, respectively. *CcGLR2.4* gives rise to CcGLR2.5 and CcGLR2.7 to CcGLR2.8. Finally, CcGLR2.8, through duplication, gives rise to CcGLR2.9 (Figure 3, Table 2).

The most recent duplication event was in family IV, where CcGLR4.4 has an ortholog in *Aquilegia coerulea*. This sequence, through duplication, gave rise to CcGLR4.3 (Figure 3, Table 2).

### 2.4. Orthologs of CcGLRs Present in Different Species

The orthology analysis using the 17 CcGLRs sequences and the annotations of 28 species under study showed that 27 species share sequences orthologous to the CcGLRs of habanero pepper. Only *Ginkgo biloba* did not show the presence of orthologs. The total number of sequences orthologous to the CcGLRs identified was 408. The four species with the highest number of orthologs were *G. hirsutum*, *Nicotiana tabacum*, *Nelumbo nucifera*, and *P. bretchneideri* with 44, 40, 23, and 19 sequences, respectively. The lowest number was presented for *M. polymorpha*, *Selaginella moellendorffii*, *Adiantum capillus-veneris*, and *P. patents* with a single orthologous sequence per species (Figure 3).

When analyzing the number of orthologs per family (II, III, and IV) (Figure 3), the most ancestral orthologs belonged to clade number III, and, specifically, one ortholog emerged to *CcGLR3.3* in *M. polymorpha* and one to *CcGLR3.5* in *P. patents*. Later, an ortholog to CcGLR3.2 appears in primitive angiosperm *A. trichopoda*, and, finally, an ortholog to CcGLR3.1 in *C. micranthum* (Figure 3).

Family II appears later than family III in evolution because the first ortholog of CcGLR2.6 was identified in *Cycas panzhihuaensis*. The CcGLR2.2 ortholog appeared later in the evolutionary timeline in *A. coerulea*. All other orthologous members diversified in large numbers in *N. tabacum* (Figure 3).

Recently, in evolution, family IV is identified through the ortholog of CcGLR4.2 in *A. trichopoda*; later, the ortholog of CcGLR4.4 appears in *A. coerulea*, followed by the ortholog of CcGLR4.3 in *M. truncatula*, and, finally, the orthologue of CcGLR4.1 in a Solanaceae (*N. tabacum*) (Figure 3).

The same ortholog pairs obtained using Orthovenn were used to obtain the rate of synonymous and non-synonymous mutations (Ka/Ks). The results showed that, despite the high selective pressure to which the GLR protein family has been exposed, all pairs of orthologs to the *CcGLRs* present values of Ka/Ks below 1 (Figure 4, Table 2), which points to the fact that the protein’s tendency is to maintain function. The orthologous sequences with the lowest Ka/Ks values are the Solanaceae under study (*N. tabacum*, *S. tuberosum*, *S. lycopersicum*). On the other hand, the species with the highest mutation rate were *Vitis vinifera*, *C. panzhihuaensis*, *F. vesca*, and *A. coerulea*.

### 2.5. Phylogenetic Analysis of CcGLRs Proteins

The CcGLR proteins were classified and named in three different clades (clades IV, II, and III), similar to the classification of GLR proteins in other reported species. Clade I included the GLRs of two species of the Malvides taxonomic group, *A. thaliana* and *G. hirsutum*, as well as the Fabides, which comprise four woody species. However, in this clade, no members belonging to the asterids and monocotyledons were found. Four CcGLR proteins were grouped in a separate clade, named clade IV, along with members of family IV of woody species, family II of *M. truncatula*, and family I of *S. lycopersicum* (Figure 5). Nine members of CcGLRs were grouped into clade II along with the other species (Figure 5). In relation to clade III, it was observed that it is shared among all species.

### 2.6. Transcription Pattern of CcGLRs

To examine whether the phylogenetic analysis could correlate with the expression profiles, in a first approach the transcriptional patterns in the leaf, stem, and root of the 17 CcGLRs genes were determined by endpoint RT-PCR, using tissue from 30-day-old habanero pepper seedlings after germination. Transcript levels were observed in 15 of the 17 CcGLRs; 2 members of clade II, CcGLR2.1 and CcGLR2.7, were the only GLRs in which no expression was observed in any of the tissues. In general, the transcript levels of family IV members were higher in stem and root, compared to those in leaves. On the other hand, the behavior for the transcript levels of the members of clade II was divided: the transcript levels of CcGLR2.2–2.4 were higher in leaf and stem than in the root, and the opposite occurred for CcGLR2.5 and CcGLR2.8, while those of CcGLR2.6 and CcGLR 2.9 were majority in root and stem, respectively. For clade III, the transcript levels of two members, CcGLR3.3 and CcGLR3.5, were similar for the three tissues studied, while those of CcGLR3.1 were more abundant in leaf and stem than in root, contrary to the behavior of CcGLR3.2 (Figure 6).

### 2.7. Molecular Docking Analysis

With the aim of identifying possible selectivity in the interaction with ligands between different CcGLRs that were evolutionarily and structurally distant, 11 CcGLRs were selected, 3 belonging to clade IV, 5 to clade II, and 3 from clade III, to carry out a molecular docking study. In this study, three widely studied ligands of these receptors were employed: D-Ser, L-Glu, and Gly. Likewise, L-Trp was incorporated as a negative control, since it was previously observed that this AA is not a GLR agonist [32], although it has been shown that it could be so in some systems [29].

Appendix A shows a representation of the molecular dockings between the ligands D-Ser, L-Glu, Gly, and L-Trp against the CcGLRs proteins represented with molecular surface diagrams and ribbons, as well as the AA interactions. As for the negative control, it was obtained that in 8 of the 11 CcGLRs tested, L-Trp was not located in the pocket reported for agonists in these proteins, its presence being observed on the surface of these proteins. However, for CcGLR4.1 and CcGLR4.2, it seemed to be partially located there, and in CcGLR2.5, this AA was in the correct site of the agonists (Appendix A).

The three AAs used as agonists in this study were in the agonist-binding pocket in all three family IV CcGLRs tested. On the other hand, for family II CcGLRs, L-Glu could not be in the pocket that corresponded to *CcGLR2.6* or *CcGLR2.9*; in both cases, it was observed on the surface of the proteins after coupling. In the case of family III of CcGLRs, only CcGLR3.3 allowed the three agonists tested to be placed correctly in the pocket, unlike CcGLR3.2 and CcGLR3.5, in which D-Ser and L-Glu, in the first case, and L-Glu, in the second, could not do so in this type of study (Appendix A).

Although in the vast majority of CcGLRs D-Ser, L-Glu, and Gly were correctly placed in the agonist binding site, the type of interactions and the AA residues that interacted with the ligand in the pocket varied, depending on the ligand and of the CcGLR (Appendix A). To go deeper into this response, some CcGLR members from each family were selected and the electrostatic potential of the surface was determined since it is a parameter that is taken into account and influences molecular docking.

The results showed important differences between CcGLR, particularly at the site that corresponds to the entry of the agonists into the pocket, as well as the pocket itself. For example, although these regions can be neutral or partially negatively charged, as is the case with CcGLR4.2, in other cases, it is observed that it is strongly negatively charged on one of the surfaces of the protein, as is the case for CcGLR3.2 and CcGLR3.5 (Appendix A).

To compare the pocket region of these CcGLRs in more detail, the data obtained from molecular docking were filtered, selecting those residues that make up the ligand-binding pocket and that were at a distance less than or equal to 12 Å. The results obtained were visualized in Pymol (Figure 7).

The results showed that several AA residues are conserved in the pocket of all CcGLRs compared to those of *A. thaliana* AtGLR, especially those of AtGLR3.3 [32]. For example, Asp81, corresponding to Asp526/532/493/495/559/559/547 in CcGLR4.2/CcGLR4.4/CcGLR2.6/CcGLR2.9/CcGLR3.2/CcGLR3.3/CcGLR3.5, respectively, and Arg88 corresponding to Arg533/539/500/502/566/566/554 for the CcGLR mentioned above. However, important changes were observed in them. For example, although a Tyr like Tyr63 of ATGLR was conserved in most CcGLRs (Tyr508/514/475/477/541) in CcGLR4.2/CcGLR4.4/CcGLR2.6/CcGLR2.9/CcGLR3.3, respectively, it was replaced by Phe541 and Asp529 in CcGLR3.2 and CcGLR3.5, respectively (Figure 7).

### 2.8. Protein–Protein Interaction

Protein–protein interaction prediction was carried out by in silico analysis of the 17 CcGLRs using the online program STRING. A total of 32 proteins were obtained that interacted with the 12 different CcGLRs present in the interaction networks. From this, these interactions were filtered, selecting only those that had been demonstrated experimentally; visualization was carried out with the Cytoscape program.

In total, six networks were obtained; one of them involved five CcGLRs. In the first network, CcGLR2.6 interacts with glucose transporters, protein kinases, 14-3-3 proteins, proteins that are part of the nuclear pore complex, MLO (Mildew Resistance Locus O) proteins, and the glutamate receptor, while the subunits CcGLR3.1, CcGLR3.2, CcGLR3.3, and CcGLR3.4 interact with the glutamate receptor (Figure 8A). A second network involved CcGLR2.3 interacting with cornichon proteins, guanylate kinase, and Ca^2+^⁺/calmodulin kinases (Figure 8B). The third network predicted interactions between CcGLR2.1 and CcGLR2.2, which interact with guanylate kinases, guanine-binding proteins, and the transcription factor HY5 (Figure 8C). The fourth network involves the protein CcGLR2.5, which interacts with pollen allergens CML25, presenilin, and guanine-binding proteins (Figure 8D). The fifth network is constituted by CcGLR2.4, which interacts with guanylate cyclases, kinesins, and GTP-binding proteins (Figure 8E). Finally, the sixth network only predicts the interaction of CcGLR3.5 with a protein containing the PBPe domain (Figure 8F).

Regarding interactions with other proteins, the results show proteins that are very active in the interaction with receptors. The guanylate kinase protein was identified in three different networks interacting five times with CcGLR2.1, CcGLR2.2, CcGLR2.3, and CcGLR2.4. Similarly, the guanine binding protein was present in three networks, although, in this case, interacting four times with CcGLR2.1, CcGLR2.2, CcGLR2.4, and CcGLR2.5. In the interaction between GLRs, two networks were observed that have the proteins CcGLR2.1 or CcGLR2.6 as a central node, interacting directly or indirectly with CcGLR2.2, CcGLR3.1, CcGLR3.2, CcGLR3.3, and CcGLR3.4. The remaining proteins were only identified in an interaction network; however, it is noteworthy that the 14.3.3 and cornichon proteins, although they were present in a single network, interacted three and five times in said networks, respectively.

With the aim of going deeper into this study and being able to corroborate that the interaction networks found could also occur in habanero pepper, a search was carried out for proteins orthologous in habanero pepper to those present in the interaction network with the CcGLR. Of the 32 sequences provided by STRING and that were used as a template, only 20 orthologous sequences were identified in habanero pepper (Table 3). Added to the above, as shown in Table 3, the AA length of the proteins that were used as a template and the orthologous habanero pepper proteins are similar. Comparing the 20 habanero pepper orthologous sequences with the template sequences, 12 show the same length and the remaining 8 proteins show minimal differences in the same aspect, suggesting the fact that they contain the same functions in the plant species under study. Among them are proteins such as cornichon, guanylate kinase, glucose transporter, protein 14.3.3, and guanine nucleotide-binding proteins, among others (Table 3).

## 3. Discussion

### 3.1. Identification of CcGLR

A total of 17 CcGLRs sequences were found in the habanero pepper genome, a number relatively like other species such *as A. thaliana* (20 AtGLRs) [2,3], *O. sativa* (24 OsGLRs) [5,6], *Z. mays* (16 ZmGLRs) [9], and *S. lycopersicum* (13 SlGLRs) [4]. However, in other species, a greater number of these proteins have been identified, such as in *M. truncatula* (29 MtGLR) [7] and other species where between 30 and 40 members have been detected, such as woody ones [8] and *G. hirsutum* (36 GhGLRs) [10].

### 3.2. Structure of CcGLRs Proteins

The results presented here confirmed that the 17 CcGLRs presented the eight characteristic domains of GLR proteins: the NTD, the LBD containing both S1 and S2, the four TMDs (M1, M2, M3, and M4), and the CTD. However, significant differences were detected between domains.

The CcGLR2.5 and CcGLR2.7 proteins presented a longer NTD, compared to the rest of the proteins. However, this region presented a very low identity between both (36%), suggesting that its additional length would not be related to the same function for these proteins. In fact, it stands out that this region is enriched in proline residues in CcGLR2.5, but not in CcGLR2.7. The NTD region of the iGluR controls various functions of the receptor, such as the assembly of the various subunits, translocation to the membrane, channel opening, agonist potency, and allosteric modulation [57,58]; however, this domain has been less studied in plants. It has been reported to participate in the binding of glutathione in AtGLR3.4 [34]. The role of these changes in NTD for the functioning of CcGLR proteins should be addressed in subsequent studies.

A more detailed analysis was performed to characterize CcGLRs motifs and identify conserved AA residues. The cation selectivity filter is determined by the residues in the M2 region [59]; the conversion of Gln (Q621), an uncharged polar AA, to Arg (R621), a positively charged polar AA, in GluA2, results in decreased Ca^2+^ permeability [60]. This site, called Q/R/N586, overlaps with the potassium channel selectivity filter of *Streptomyces lividans* (KcsA, accession number: PIR S60172) and the glutamate receptor GluR0 of the cyanobacterium *Synechocystis* sp. (SynGluR0, accession number: BAA17851) [61], reinforcing the role of this domain in determining ionic permeability.

All CcGLRs possessed a Phe in this position, except CcGLR3.1 and CcGLR3.5, in which it is modified to a Lys and Thr, respectively. The majority presence of Phe in that position seems to be a generality in plant GLRs. For example, 16 of the 20 AtGLRs possess this residue [62]. Experimental evidence recording the permeability of these proteins to different cations is scarce and is lacking in species other than *A. thaliana*. The complexity regarding the formation of hetero or homotetramer for the formation of the functional receptor when GLRs are expressed in heterologous systems, as well as the lack of knowledge of the profiles of their agonists, have contributed to this problem [15].

However, Ca^2+^ conduction in GLRs containing a Phe residue at this position has been demonstrated for the *A. thaliana* proteins AtGLR3.4 [21] and AtGLR1.4 [29], and of rice OsGLR2.1 [63]. Likewise, AtGLR3.5, which presents a Gly instead of Phe, also appears to conduct Ca^2+^ currents, but in the mitochondrial inner membrane [64], and AtGLR3.7, which changes this Phe for a Lys, induced a for Ba^2+^, Ca^2+^, and N^a+^ [14]. Studies that address the impact on the ionic selectivity of the changes in this residue of the GLRs that are indicated in this work, as well as those previously reported for other species, are necessary to determine the function of these proteins.

Interestingly, the presence of a P in position 11 of the M2 domain alignment is highlighted, which was exclusive for the CcGLR2.7–2.9 proteins. This P was not present in the GLRs of tomato, the species closest to habanero pepper, where these proteins have been previously studied, nor in the GLRs of *A. thaliana*. It is known that P residues are frequently found occurring in hinge regions where they promote domain-swapped structures, although this function is highly context-dependent [65].

It is known that the SYTAXLXXX domain in M3 is highly conserved, named the gate region, which participates in the opening of the channel, once the ligand binds to the LBD region [66]. The presence of an S in position eigh in all CcGLRs could lead to channel activity, even in the absence of the ligand, as has been demonstrated for AtGLR3.2 and AtGLR3.3 of *A. thaliana* [20], and that of the moss PpGLR1 [67]. Ref. [68] demonstrated that changing an A, completely conserved at this position in iGluRs, for a T, a polar hydrophilic AA like the conserved S in most plant GLRs, results in a constitutively open channel.

All CcGLRs also presented an A in position 4 of this domain. It has been shown that this residue is the most conserved and is present in all mammalian, plant, and bacterial receptors. Its role is to act as the hinge that pulls the M3 helix away from the others, leading to the opening of the channel [66]. Likewise, T in the third and seventh positions of this motif were conserved in 100% and 65% of the CcGLRs, respectively. These residues are part of a narrow constriction of the pore [37]. The T at position seven was modified to S in the receptors CcGLR2.7–2.9 and CcGLR4.1, CcGLR4.3–4.4, which has been previously reported for members of clade II *A. thaliana* [69].

The distinctive character of the residues forming the gate region in M3, as well as the selectivity filter in M2 in the CcGLRs of clade IV (CcGLR4.1, CcGLR4.3, and CcGLR4.4) and clade II (CcGLR2.7–2.9) suggest that the opening and permeability properties of these proteins could be different from those of AtGLRs, and even tomato SlGLRs. Based on our results, it is interesting to propose these proteins as candidates for studies of the ion selectivity and permeability of the plant GLRs, with the aim of expanding the knowledge of the function of the M2 and M3 regions in these processes.

Despite the discovery of GLRs in the *A. thaliana* genome more than two decades ago [2], and recently in other species [3,4,5,7,8,9,10,70,71], the subcellular localization and processing in the secretory pathway has not been able to be elucidated for most of them. Localization to the plasma membrane has been demonstrated only for seven AtGLRs, AtGLR1.2 [27], AtGLR1.3 [72], AtGLR1.4 [29], AtGLR3.2 and AtGLR3.4 [15,25,73], AtGLR3.1 [20], AtGLR3.7 [14]; one OsGLR, OsGLR2.3 [63]; one PbrGLR, PbrGLR3.3 [74]; and one *Raphanus sativus* RsGluR [75]. Furthermore, it has been shown that AtGLR3.4 can be localized to the chloroplast membrane [76] and one splicing variant of chloroplast [64], whereas AtGLR2.1, located in the complex vacuolar system [20], and AtGLR3.1, AtGLR3.3 [77], OsGLR1.1 [5], and OsGLR3.1 [6], located in endoplasmic reticulum. However, the localization of GLRs belonging to the Solanaceae family is not yet known.

Interestingly, CcGLR2.5 was the only CcGLR with a predicted subcellular localization to the nucleus, in addition to the plasma membrane. This protein apparently does not present a signal peptide at the N-terminus, an exclusive characteristic for family II of CcGLRs, except for CcGLR2.7 and CcGLR2.8. It has been suggested that the presence of a signal peptide in iGluRs is important for the nascent protein to enter the secretory pathway [78]. It has been predicted that 16 of the 20 AtGLRs contain this signal peptide at the N-terminus, just as it happens for all members of CcGLRs from families III and IV. Of the AtGLRs that present a low probability for the presence of this signal peptide, it is interesting that two of them present predicted (AtGLR2.5) or documented (AtGLR3.5) localization outside the secretory life, namely in the mitochondria and/or chloroplast [64]. Likewise, a third is AtGLR3.3 [77], which is also not located in the plasma membrane.

These previous data allow us to suggest that, in addition to CcGLR2.5, the rest of the CcGLRs belonging to clade II, whose presence of a signal peptide in the N-terminus could not be predicted, could not enter the secretory pathway, and be in other organelles more than in the PM. However, the experimental data that exist are inconclusive. C-terminal-dependent protein–protein interaction has also been a mechanism to regulate the subcellular localization of GLRs [20]. Also, interestingly, CcGLR3.3 was the only CcGLR protein that exhibited NLS localized to the CTD, like its *A. thaliana* ortholog AtGLR3.3. This signal is essential for cell–cell communication in response to wounding [79], suggesting that CcGLR3.3 could function in a similar way in habanero pepper, although the functionality of this motif must be demonstrated experimentally. Despite the usefulness of in silico analyses, efforts must be directed to know the precise subcellular localization of these proteins to elucidate their function in plants.

Recycling and degradation of receptors need to be properly targeted and protein phosphorylation appears to regulate this purpose, in addition to many cellular processes such as protein activity, localization, and mobility [11]. It has been reported that the phosphorylation of GluRs is a key mechanism that regulates the function of ion channels and occurs in the C-terminal region of the receptors [11,80]. In this work, the accumulation of possible phosphorylation sites in the CTD of these proteins was evident, and this accumulation was differential depending on the type of CcGLR. The repercussions of these differences on their function must be studied.

The majority of CcGLRs show a genetic structure of five exons and four introns (47%). About 50% of *A. thaliana* AtGLRs have six exons and five introns and the other approximately 50% contain five exons and four introns; however, it also contains a sequence containing eight exons and seven introns [62]. In the woody *species P. bretchneideri*, the genetic structure is also maintained. Of the 34 PbrGLR sequences, 64% have the structure of five exons and four introns [8]. Comparing these results between these distant plant species, it can be suggested that the structure of the genes has been maintained throughout evolution.

The physical location of *CcGLRs* is also an important element to consider to know the history of GLRs. Ten of the 17 CcGLRs sequences were grouped in tandem on three different chromosomes (3, 4, and 6), representing 58% of the sequences. In *A. thaliana*, 55% of the AtGLRs genes (10 sequences) are grouped in tandem on chromosomes two and five [62]. In *S. lycopersicum*, of the 13 *SlGLRs* sequences, 11 are grouped in tandem (84%) on chromosomes 2, 4, 6, and 7 [4]. In *S. spontaneum*, 21 *SsGLR* sequences out of 34 are found clustered in tandem (62%) on chromosomes 2A, 2B, 2C, 4A, 4C, and 8C. The closeness that plant GLR sequences maintain is interesting and suggests the possible evolutionary force that gave rise to the presence of more GLR members.

This is corroborated by the results of the habanero pepper duplication events, where *CcGLR1.3* and *CcGLR1.4* of family I are found in tandem, CcGLR2.1, CcGLR2.2, CcGLR2.3, CcGLR2.4 and CcGLR2.5 of family II, and with the greater number of members, are also located in tandem and, finally, *CcGLR3.1* and *CcGLR3.2* of family III are grouped in the same way. On the other hand, two segmental duplication events occurred, with *CcGLR2.2* and *CcGLR2.7* located on chromosomes 6 and 3, respectively, and *CcGLR3.3* and *CcGLR3.5*, on chromosomes 4 and 2. This evolutionary force is very important because it suggests the existence of a strategy used by plant species as part of adaptive processes [81]. However, according to previous results, the expansion of GLRs is focused on tandem duplication events, which agrees with research carried out in *F. vesca*, *P. mume*, and *P. persica*, where the tandem duplication event is the majority, with 61%, 61%, and 77%, respectively [8].

In the case of family II, *CcGLR2.2* is the first sequence to be identified as an ortholog of a monocot sequence (*Z. mays*) and is responsible for giving rise to *CcGLR2.3* through tandem duplication. The duplication event is the most ancient. Later duplication events of family II were more recent and possibly occurred in the Solanaceae. *CcGLR2.3* through tandem and segmental duplication gives rise to *CcGLR2.4* and *CcGLR2.7*, respectively. *CcGLR2.4* gave rise to *CcGLR2.5* and *CcGLR2.7* to *CcGLR2.8*. Finally, *CcGLR2.8,* through duplication, gives rise to *CcGLR2.9*. The most recent duplication event is in family IV, where *CcGLR4.4* has an ortholog in *A. coerulea*. This sequence through duplication gives rise to *CcGLR4.3*.

Orthologs of family III CcGLRs are the most numerous and show an expansion in all species. Specifically, in *G. hirsutum*, 44 orthologs were identified, of which more than 80% belong to clade number III. On the other hand, the species that contain the greatest number of orthologs to the CcGLR in the three families are the Solanaceae.

Regarding the value of Ka and Ks, as observed in the results, the values of the ratio are below 1, indicating that, despite the strong pressure of evolutionary selection, these sequences have been inclined to maintain their function, as previously reported [9].

### 3.3. Phylogenetic Analysis and Transcription Patterns of CcGLRs

Phylogenetic analyses allow the classification of a family of genes within clades that may be functionally distinct. The CcGLRs were grouped into three different clades where the sequences of clade IV, represented by four genes, were shared tomato, and in other branches, there were rice sequences of clade III and IV, as well as members of clade IV of the woody species. This suggests the divergence of a clade IV distinct from that of *A. thaliana*, a dicotyledonous species member of the Brassicaceae family, and rice, a monocotyledonous species member of the Poaceae family, whose members were grouped in a clade different from that of Solanaceae, habanero pepper, and tomato (both dicotyledonous species).

Two of the habanero pepper clades (clade II and III) were homologous to those of all species used in this analysis. This result suggests that these two clades evolved in a land plant ancestor before the divergence of the Solanaceae and Brassicaceae families. In addition, the nine members of family II of *A. thaliana* share a specific arm with the GLR2.6 members of tomato and habanero pepper, and the other members of said clade are grouped in close arms within the same clade (CcGLR2.1-CcGLR2.5 and CcGLR2.7-CcGLR2.9). These genes are encoded in tandem on chromosome 6, while CcGLR2.6 is encoded on chromosome 8. These data suggest that CcGLR2.6 is an ancestral clade II gene, and other members of clade II increased gene number because of gene duplication.

Based on the results of transcript profiles, the results seem to suggest that there is no clear correlation between these and the phylogenetic clade to which they belong; only in clade IV were those members grouped that were mainly expressed in stem and root. Unfortunately, no ortholog of CcGLRs from clade IV has been functionally characterized in other species, so the role of these proteins in those plants that possess it is still unknown. The expression of the only two orthologs reported in tomatoes, *SlGLR1.1* and *SlGLR1.2*, coincides with our results, and both genes are expressed mainly in the root of tomato vegetative organs [4].

Interestingly, transcripts for CcGLR2.1 and CcGLR2.7 were not observed in any of the tissues evaluated, suggesting that these genes are not expressed in habanero peppers. If this were the case, these could be pseudogenes, a common process in biological systems due to transposon duplication or insertion events, which lead to gene inactivation [82]. As shown in this work, both genes were obtained from tandem duplication events in habanero pepper. Likewise, no considerable expression levels were detected for *SlGLR2.1*, an ortholog of CcGLR2.1, through real-time RT-PCR, in tomato vegetative and reproductive organs. In the case of CcGLR2.7, as reported in this work, only one ortholog to this protein was found in the Solanacea *N. tabacum*, a species in which GLRs have not yet been studied. In any case, it would be important to evaluate the expression of these two genes under different developmental conditions, response to different environmental stresses, as well as in other tissues (floral organs and fruits).

The expression results presented here for *CcGLR2.2* and *CcGLR3.1* agree with those reported by [4] for the orthologs in tomato, SlGLR2.2 and SlGLR3.1, with a majority expression in leaf and stem, but not for the ortholog to CcGLR2.6, in which these authors report a non-considerable expression in tomato. These results suggest that, although some orthologous proteins could function similarly between these two evolutionarily close species, this is not always the case, and these discrepancies must be resolved in subsequent studies.

Only two genes, *CcGLR3.3* and *CcGLR3.5*, were expressed similarly in all tissues. Interestingly, the orthologs of these genes in *A. thaliana*, AtGLR3.3, and AtGLR3.6, respectively, have been considered synergistic master regulators, participating in the long-distance response by mechanical wounding [77,83]. The orthologs of these two genes in tomato have been the only ones whose function has been studied in more detail in this species. Both genes increase their expression during cold acclimation to 12 °C of tomato plants, leading to an increase in tolerance when they are then exposed to 4 °C. This tolerance seems to be given by a regulation of apoplastic H_2_O_2_ production and redox homeostasis [55]. The next challenge would be to investigate whether this function is conserved for its counterpart in habanero pepper.

### 3.4. Molecular Docking

The results previously described in this section provide circumstantial evidence of the interaction of GLRs from habanero pepper and three ligands, glutamate, glycine, and D-Ser. The shell architecture reported by all LBD of GLR of plants [84] was also conserved to CcGLRs.

The crystallized structures of the LBD domains in complex with different agonist amino acid AAs from AtGLR3.2–3.4 [32,36,37], showed equivalent coordination for each agonist, allowing the establishment of a consensus motif that coordinates the carboxyl and amino groups of the AAs, Asp-Ala/Thr-Arg-Phe/Tyr-Glu-Tyr. Of these 6 residues, 3 were conserved for all 11 CcGLRs studied through molecular docking, which were Asp-X-Arg-Phe-X-X. Interestingly, all clade III CcGLRs conserved 100% of these residues (Asp-Ala/Thr-Arg-Phe-Glu-Tyr). These results suggest that CcGLR clade III members can coordinate the amino and carboxyl groups of AAs, in a manner like those of *A. thaliana*. Although these residues were conserved in all CcGLRs of clade III, only in CcGLR3.3 were all these residues positioned in the pocket, close to the agonist, showing the greatest number of interactions, like AtGLR3.3 from *A. thaliana* [32]. In contrast, CcGLRs CcGLR2.6, CcGLR2.9, CcGLR3.2, and CcGLR3.5 did not allow Glu entry into the pocket.

Important differences were observed in the agonist binding pocket. For example, CcGLR2.6 lacks the Arg that has been reported to provide a key interaction, stabilizing the AA side chain [85], which is also absent, forming the pocket of CcGLR3.2, CcGLR3.5, and members of clade IV. This residue is modified to a Lys in CcGLR2.9. In all cases in which the agonist Glu does not enter the pocket, it is observed that the electrostatic potential of the surface, close to the entrance of the pocket, is highly negative, suggesting that the entry of the AA Glu could be difficult. A valine substitution for the conserved Glu was observed in the pocket of CcGLR2.9 and CcGLR4.2.

The CcGLRs that were least conserved in these residues participating in the coordination of amino and carboxyl groups were CcGLR4.2 and CcGLR2.7, which changed Ala to Asp and Glu to Val. However, in the molecular docking analyses, the three agonists tested were in the pocket of these proteins. These results suggest that, although the coordination of the carboxyl and amino groups of AAs can occur, the amino acid residues participating in them can vary in these receptors.

These changes in the amino acid residues forming the pocket between the CcGLR components suggest that versatility may occur in terms of binding with different agonists, as well as the affinity with which they bind to the receptor, as has been reported for the AtGLRs [29,63,86]. The presence of a CcGLR battery in the habanero pepper roots, which perceive fluctuating environments of N-amino acids, could be important for the development and productivity of this species. The physiological relevance of these changes in the CcGLR should be evaluated in subsequent studies. However, although molecular docking to date has been very useful, for example, for the design and discovery of new drugs, it cannot be ignored that this methodology has limitations that must be taken into account, for example, the lack of synergistic computational models, quality database, and standardization to test and validate the results.

### 3.5. Protein–Protein Interaction

Binding of receptors to a variety of proteins is known to regulate their targeting to the synapse and, consequently, synaptic strength is modulated, and modification of receptor characteristics is possible. In this study, the 17 CcGLRs predicted interaction with different proteins and were grouped into six different interaction networks.

The first network comprised the interaction of CcGLR2.6 with regulatory proteins (such as 14-3-3 proteins), transport molecules (such as glucose transporters), and calcium-dependent kinases. Furthermore, this same network involved the interaction of CcGLR2.6 and CcGLR3.1–3.4 with glutamate receptors. The interaction of five GLRs of *A. thaliana* (AtGLR1.2, 2.1, 2.9, 3.4, and 3.7) with 14-3-3 proteins has been identified by affinity chromatography [17]. In this same species, AtGLR3.7 was found to interact with 14-3-3 proteins, participating in the response to salt stress in *A. thaliana* by affecting calcium signaling pathways [87]. Furthermore, the interaction of some GLR subunits that can interact with multiple other GLRs (AtGLR1.1, AtGLR2.9, AtGLR3.2, and AtGLR3.4) has been demonstrated using a two-hybrid system [13]. The ortholog of CcGLR2.6 in tomato (SlGLR2.6) has not yet been functionally characterized. In *A. thaliana*, this ortholog (AtGLR2.8) participates in the root response to wounding [88].

The second network, constituted by CcGLR2.3, predicted interactions with cornichon proteins, myosin proteins, and guanylate kinases. The interaction, through imaging assays, of GLR proteins with cornichon proteins was demonstrated in *Caenorhabditis elegans*, and their association is involved in the trafficking of GLRs [89]; in addition, cornichon proteins are capable of associating with GLRs to direct them to the plasma membrane [20]. Interestingly, in this work, we could not predict the presence of a signal peptide in the N-terminus for CcGLR2.3, which is important for entering the secretory pathway. The possible interaction of this CcGLR with cornichon proteins could be important for its translocation to the PM, which must be demonstrated experimentally.

On the other hand, the interaction of Discs large (DLG), which is a protein containing multiple PDZ domains that belong to the family of molecular scaffolding proteins known as membrane guanylate kinases, with glutamate receptors in *R. norvegicus* [90]. In addition to this, interactions between iGluR-NMDA with yeast guanylate kinase have been recognized, postulating that their association is important for structural organization [91].

The third network shows the prediction of CcGLR2.1 and CcGLR2.2 with binding proteins, guanylate kinases, and guanine nucleotide proteins. Gnb1 and Gnb2 are beta subunits of guanine nucleotide-binding protein (G proteins) and their interaction with NMDA in *Mus musculus* [92]. Guanylate protein kinases, specifically Dlg4, have also been shown to interact with some NMDA receptor subunits in *M. musculus* and are required for synaptic plasticity associated with NMDA receptor signaling [93].

A fourth network is constituted by the interaction of CcGLR2.5 with binding proteins and presenilin. Calmodulin and calcium-binding proteins, interact with proteins with the NMDA receptor of *M. musculus*, specifically the NR2 and NR1 subunits, respectively [94]. The interaction of the GluR1 subunit of the *R. norvegicus* AMPA receptor with guanine-binding proteins has also been demonstrated [95]. Presenilin (PS1) interacts with NMDA suggesting a stable complex between the two that gives rise to the correct localization or synaptic release of the receptor [96].

Similar to the previous network, a fifth network involved the CcGLR2.4 with binding proteins and guanylate kinases. The interaction between glutamate receptors and guanosine triphosphate (GTP)-binding protein was demonstrated in *R. norvegicus*, allowing highly localized signaling by cyclic AMP in neurons [95]. On the other hand, the interaction with guanylate protein kinases was evidenced in *M. musculus* [93] and *R. norvegicus* [90].

Finally, a sixth network involves the prediction of the interaction of CcGLR3.5 with PBPe domain-containing protein. This protein appears to be a fragment of an as-yet uncharacterized GLR-like protein. In the literature, we found that AtGLR3.4 interacts with other GLR subunits of *A. thaliana* [13], as we previously mentioned, and, in addition, the physical interaction of AtGLR3.2 and AtGLR3.4 between themselves was demonstrated through FRET performed in the kidney (HEK) and *Nicotiana benthamiana* leaf cells [15]. In addition to this, the heteromeric interaction of these last two GLRs (AtGLR3.2 and AtGLR3.4) activated by 1-aminocyclopropane-1-carboxylic acid (ACC) is suggested, different from this, apparently AtGLR3.3 and 3.6 do not they assemble into a heteromeric complex [97].

Protein–protein interaction networks are very important, as these modules are the ones that determine the functions of proteins in many cases. Unfortunately, very few interactions of plant GLRs with other proteins have been functionally characterized. In this work, no protein interaction could be detected with members of family IV, nor with those of clade II CcGLR2.7, CcGLR2.8, and CcGLR2.9. This result can be explained because to date this interaction has not been characterized for any orthologue of these proteins, nor have they been functionally characterized. On the other hand, our results suggest that for the functioning of the rest of the members of clade II, the interaction with other proteins could be important, including, for example, for their subcellular localization, signaling cascade regulated by protein kinases and calcium, and formation of a receptor functional through its union with clade III proteins, among others.

The detection in the habanero pepper genome of proteins orthologous to those predicted to interact with CcGLRs, as well as the exclusive use in this work of interactions demonstrated experimentally, contribute to strengthening the possible physiological occurrence of the interactions predicted here. However, future studies should be directed to experimentally demonstrate these interactions of CcGLR proteins to understand their function in habanero pepper.

## 4. Materials and Methods

### 4.1. Identification and Annotation of GLR Family Members in Habanero Pepper

To identify *CcGLR sequences*, different strategies were used. First, a “BLASTP” search (default parameter) was performed using the BioEdit Sequence Alignment Editor software version 7.2.5 [98], which contained the file of habanero pepper coding sequences previously downloaded from the National Center for Biotechnology Information database (https://www.ncbi.nlm.nih.gov/datasets/genome/?taxon=80379 (accessed on 26 March 2021). The 13 *S. lycopersicum* GLR protein sequences (SlGLR) [4] were retrieved from the National Center for Biotechnology Information’s (NCBI, https://ncbi.nlm.nih.gov/ (accessed on 30 November 2016)) database (Appendix A) and used as a query in this analysis, using default parameters. Secondly, the “GluR_Plant” domains of tomato GLRs [4] were used as a query in a “BLASTP” analysis through the NCBI platform (https://blast.ncbi.nlm.nih.gov/Blast.cgi (accessed on 12 April 2021)).

Furthermore, the sequences obtained were filtered following the following criteria: e-value of 1.0 × 10^−100^, presence of the GluR_Plant domains, ligand binding (LBD, S1 and S2) and transmembrane domains (DTM, TM1-TM4). Transmembrane regions were predicted by using the bioinformatics program TMPred (https://embnet.vital-it.ch/software/TMPRED_form.html (accessed on 13 March 2023)) and TMHMM version 2.0 [99] (http://www.cbs.dtu.dk/services/TMHMM/ (accessed on 13 March 2023)), while the rest of the domains were identified through the NCBI Conserved Domain Search (https://www.ncbi.nlm.nih.gov/Structure/cdd/wrpsb.cgi (accessed on 13 March 2023)). Those sequences that meet these criteria were retained for further analysis as CcGLR.

### 4.2. Structural Analysis and Main Characteristics of the CcGLR Family

To verify the probable start and end sites of each domain, multiple alignments were performed using default parameters of Clustal Omega version 1.2.4 ((https://www.ebi.ac.uk/Tools/msa/clustalo/ (accessed on 29 November 2023) and the *A. thaliana* [3]) sequences as the query sequences. The alignments of the LBD region were performed with the online tool T-COFFEE [100] (https://tcoffee.crg.eu/apps/tcoffee/do:expresso (accessed on 29 November 2023)), using as a template the structural information of the LBD domain of AtGLR3.3 [32]. The domains were visualized in Jalview version 2.11.2.6 [101].

The predicted molecular mass and isoelectric point (pI) were obtained from Protparam (https://www.expasy.org/resources/protparam (accessed on 20 July 2021)).

To predict the subcellular localization of proteins, three bioinformatics tools were used: DeepLoc-1.0 [102] (http://www.cbs.dtu.dk/services/DeepLoc/ (accessed on 4 September 2023)), WoLF PSORT [103] (https://wolfpsort.hgc.jp/ (accessed on 4 September 2023)) and Plant-mPLoc [104] (http://www.csbio.sjtu.edu.cn/bioinf/plant-multi/ (accessed on 4 September 2023)). Multiple Em for Motif Elicitation (MEME, https://meme-suite.org/meme/tools/meme (accessed on 13 March 2023)) [105] was used to confirm conserved motifs in CcGLRs protein sequences and keep the default parameters.

A search for Nuclear Localization Signal (NLS) motifs in the CcGLRs sequences was performed using the online tool NLStradamus: a simple Hidden Markov Model for nuclear localization signal prediction [106] (http://www.moseslab.csb.utoronto.ca/NLStradamus/ (accessed on 7 September 2023)), introducing the amino acid sequence of each CcGLR. The prediction cutoff used was 0.5.

The presence of possible signal peptides in the CcGLRs was carried out through the online program Phobius prediction [107] (https://phobius.sbc.su.se/ (accessed on 23 August 2023)), using the default parameters.

The prediction of phosphorylations of CcGLR proteins was carried out on the MusiteDeep web server (https://www.musite.net/ (accessed on 29 June 2023)), based on deep learning [108,109,110], using the parameters granted by default by the program. The results of the possible phosphorylation sites were visualized in TBtools [111], using the text file generated in MusiteDeep.

### 4.3. Phylogenetic Analysis

A multiple alignment was performed with the full-length GLR amino acid sequences, using the ClustalW program in MEGA6 with the following parameters: gap-to-change = 10, 15, and 20; gap extension cost = 1; and amino acid substitution matrix = Blosum 30. The alignments included the GLR sequences of habanero pepper and 11 other species in which they have previously been reported, for a total of 284 sequences: 17 CcGLRs from *C. chinense*, 13 GLRs from *S. lycopersicum* (SlGLR) [4], 20 GLRs of *A. thaliana* (AtGLR) [2,3], 16 GLRs of *O. sativa* (OsGLR) [5], 15 GLRs of *M. truncatula* (MtGLR) [7], 143 GLRs of the woody plants *P. bretschneideri* (33 PbrGLR), *F. vesca* (36 FveGLR), 34 GLRs *P. mume* (34 PmuGLR) and *P. persica* (40 PpeGLR) [8], 25 *S. spontaneum* GLRs (SsGLR) [9], 28 *G. hirsutum* GLRs (GhGLRs) [10], and 5 *Z. mays* GLRs (ZmGLR) [9] (Appendix A). The GLR of the cyanobacterium *Synechocystis* sp. (SynGluR0, accession number: BAA17851) and the bacterial periplasmic amino acid binding protein GlnH (EcoliglnH, accession number: X14180) were used as external sequences. The construction of the phylogenetic tree was carried out using the Maximum Likelihood method, and the Poisson model with partial deletion. The Bootstrap values were obtained from 1000 replicate runs (MEGA6.0 software) [112].

### 4.4. Transcription Patterns of CcGLRs

Leaf, stem, and root from 30-day-old habanero pepper seedlings were used for the extraction of total RNA. These seedlings were grown under the conditions reported by [113]. Total RNA was extracted using Trizol^®^ reagent (Invitrogen, cat. 15596026), following the manufacturer’s instructions. Prior to reverse transcription, the samples were treated with the Turbo DNAse kit (2 U/µL) from Invitrogen (cat. AM2238). For the synthesis of the first strand cDNA, the first strand cDNA synthesis^®^ reagent set from ThermoFisher Scientific was used, using 1 µg of RNA, according to the manufacturer’s instructions.

For the endpoint PCR amplifications of the *CcGLRs*, the specific primers, whose characteristics are described in Appendix A, and the Taq DNA Polymerases^®^ (cat. 10342053, Invitrogen) were used. The annealing temperature for each PCR reaction is described in the table, using 32 cycles in the amplification reactions. The EF1α gene was used as a loading control, using the primers: in sense 5′-AGCGTGGTTATGTTGCCTCA–3′ and in antisense 5′-GGAAGTGTGGCAGTCGAGAA–3′ (Amplicon size of 156 bp).

### 4.5. CcGLRs Gene Structure, Chromosomal Localization, and Gene Duplication

The identification of the intron and exon sequences, as well as the location of the *CcGLR* genes on the habanero pepper chromosomes, was carried out by consulting each CcGLR in the NCBI database (https://www.ncbi.nlm.nih.gov/genome/gdv/browser/genome/?id=GCA_002271895.2 (accessed on 21 March 2023), entering the identification number of each CcGLR (Table 1). The graphical representation of the CcGLRs on the chromosomes was performed using the “Gene Location Visualize” tool of TBtools [105] and the chromosomal coordinates previously obtained for each CcGLR gene from NCBI.

For the detection of segmental and tandem duplication, the Multiplex Collinear Scanning Toolkit (MC-ScanX) was used with the default parameters [114]. Also, genes separated by less than 10 genes and/or a distance of less than 30 Kbp were considered tandem duplicated genes [62].

### 4.6. Ortholog Analysis

For the identification of orthologous proteins, the protein annotations (Appendix A) of *P. patents*, *M. polymorpha*, *S. moellendorffii*, *A. capillus-veneris*, *G. biloba*, *C. panzhihuaensis*, *A. trichopoda*, *C. micranthum*, *S. intermedia*, *D. nobile*, *O. sativa*, *H. vulgare*, *B. distachyon*, *S. spontaneum*, *Z. mays*, *N. nucifera*, *A. coerulea*, *V. vinifera*, *A. thaliana*, *G. hirsutum*, *F. vesca*, *P. bretchneideri*, *P. mume*, *P. persica*, *M. truncatula*, *C. chinense*, *S. lycopersicum*, *S. tuberosum*, and *N. tabacum* were used.

The annotation file of each species was compared with the 17 CcGLRs of habanero pepper using the OrthoVenn3 online tool (https://orthovenn3.bioinfotoolkits.net/home (accessed on 23 June 2023)) [115] using the Evalue parameters of 1 × 10^−2^ the and inflation value of 1.5. The orthologs of the CcGLRs for each plant species were shown in a heatmap with the help of the “HeatMap Illustrator” tool of the TbTools program [105].

### 4.7. Taxonomic Tree

To construct the taxonomic tree of the analyzed species, the MEGA11 program was used [116]. In this sense, a text file was constructed with a list of the names of the 29 species mentioned and submitted to the program. We proceeded to use the MEGA “Time Tree” option, which is a public database of thousands of publications related to the evolution of species [116].

### 4.8. Ka/Ks Ratio

The Ka/Ks relationship between pairs of orthologous genes was calculated using the “Simple Ka/Ks calculator” option of the TBtools program. For this analysis, the coding sequence of all CcGLRs and all their respective orthologs was used.

### 4.9. Molecular Modeling and Docking

Protein preparation: Most of the .pdb computational models of 3D structures of the CcGLRs were obtained from an AlphaFold structure prediction (https://alphafold.ebi.ac.uk/ (accessed on 31 July 2023)) (Appendix A). Subsequently, a refinement was performed on each model to exclusively isolate the segments corresponding to the ligand-binding domain (LBD) using the PyMol software (the PyMOL Molecular Graphics System, Version 2.0 Schrödinger, LLC). As an exception, the CcGLR2.7 LBD 3D structure was generated using the SWISS-MODEL [117] online server (https://swissmodel.expasy.org/ (accessed on 31 July 2023)) and the CcGLR2.8 LBD structure as a template. Files were saved in .pdb format for molecular docking analysis. To generate the GLR LBD models, the residues were used to form the S1 and S2 LBD segments (Appendix A).

Preparation of the ligands: the 2D model of the ligands L-Glutamate, Glycine and D-Serine were obtained from the PubChem platform (https://pubchem.ncbi.nlm.nih.gov/ (accessed on 09 August 2021)) and prepared for docking with the platform by Avogadro: an open-source molecular builder and visualization tool Version 1.2.0 (https://avogadro.cc/ (accessed on 31 July 2023)) [118], minimizing the energy with the MMFF94 force field and the 2D structure was saved as .mol2.

Docking study: Molecular docking of each CcGLR with the ligands was carried out in the AutoDockTools version 1.5.6 software (ADT, http://autodock.scripps.edu/downloads/resources/adt/index_html (accessed on 31 July 2023)) with AutoDock Vina (http://vina.scripps.edu/ (accessed on 31 July 2023)) [119].

Once the molecular docking models were obtained, the files with the coordinates corresponding to the ligand and the receptor were exported for visualization in the PyMol program. Considering the conserved AAs in the alignment and by increasing numbering, the conserved residues corresponding to the “core” were selected and formed by residues in S1 and S2, as reported in [32]. By approaching 12 Å of any of the AAs that make up the “core”, the models in which the ligand was found within the canonical pocket in the GLRs were selected.

### 4.10. Protein Interaction 

The prediction of CcGLR interaction with other proteins was performed with the Search Tool for the Retrieval of Interacting Genes/Proteins (STRING) (version 10.0, https://string-db.org/ (accessed on 8 October 2021)) [120], using one protein as a reference. The reference proteins to carry out the interaction prediction were selected according to the highest bitscore value obtained from the program (Appendix A). Of the total network interactions, only those interactions that have been determined experimentally were selected. The protein interaction networks were visualized in Cytoscape 3.8.2 (https://cytoscape.org/ (accessed on 15 February 2022)) [121]. The accession numbers and names of the proteins associated with the CcGLRs are listed in Appendix A.

Principio del formulario

Also, a search for orthologs in habanero pepper, using OrthoVenn3 (https://orthovenn3.bioinfotoolkits.net/home (accessed on 17 March 2023)) [115], was carried out from those proteins that were identified in the interaction networks. 

## 5. Conclusions

Although GLRs regulate key processes in plant development, these proteins have not been studied in the vast majority of plant species. This is the first study that shows an approach, not only in the identification, structural characterization, and phylogenetic analysis, but also in the prediction of possible agonists of the CcGLR of habanero pepper.

The results of this research revealed the presence of 17 sequences with high similarity to GLRs, grouped evolutionarily into three clades, and two of them, II and III, are shared with some members of *A. thaliana*, while clade IV is shared with two tomato GLR members. The results of the comparison of domains, orthology, phylogeny, and molecular docking indicated that some members of clade II and IV of the CcGLR, preferably those that arose later in evolution, present unique characteristics, even different from those of tomato, a species closer to habanero pepper and the only member of the Solanaceae previously studied. This result allows us to speculate that these proteins may have new functions, different from those already reported for *A. thaliana* and tomato. The results obtained are useful for directing deeper research, aimed at the subcellular localization of these proteins, their selectivity and ionic permeability, and identification of their possible agonists, shedding light on their function in plants.

## Figures and Tables

**Figure 1 plants-13-00812-f001:**
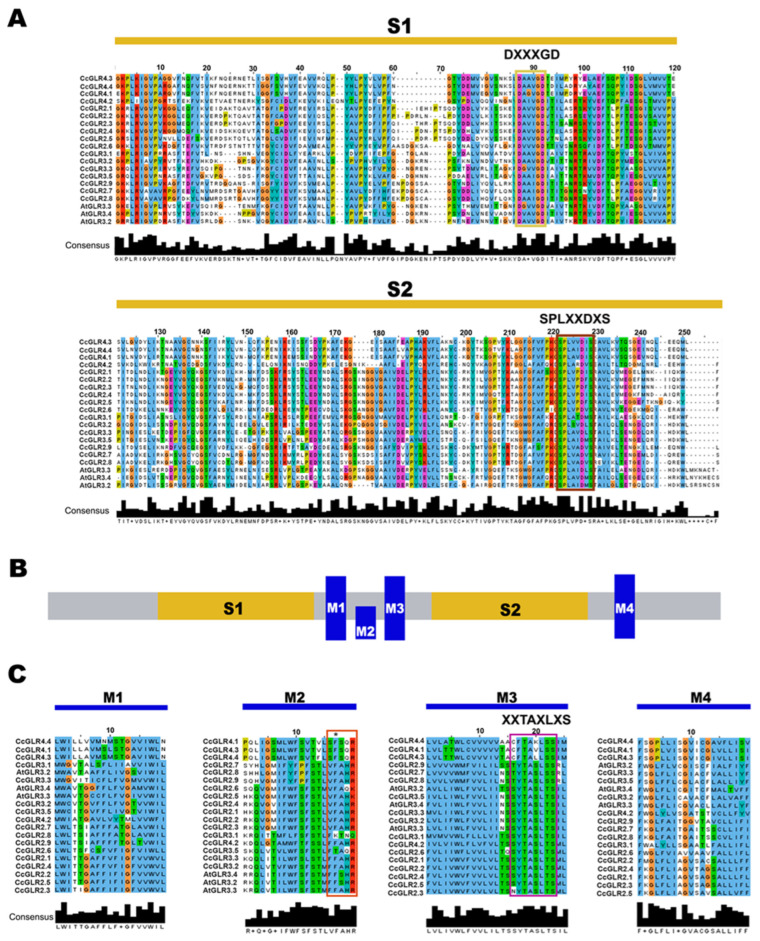
Alignment of the S1 and S2 domains (**A**) and the M1-M4 TMDs (**C**) of the CcGLRs proteins. Visualization of domain positions in CcGLR proteins is shown in (**B**). AtGLR3.2–3.4 are used as reference in alignments. DXXXGD motif in S1 and SPLXXDXS motif in S2 are indicated in yellow and red boxes, respectively. Phenylalanine conserved in selectivity filter from M2 also is marked with an asterisk. The position of selectivity filter of the K channel KcsA and the bacterial glutamate receptor GluR0 is marked by a red box in M2. XXTAXLXS domain is indicated in M3 by the pink box from M3.

**Figure 2 plants-13-00812-f002:**
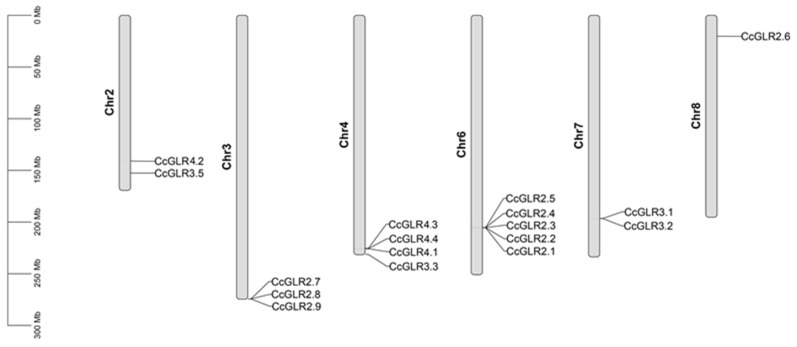
Genomic distribution of *CcGLRs* on habanero pepper chromosomes. Genetic distances are indicated in Mb on the left side of the figure, with each chromosome’s name located adjacent to it.

**Figure 3 plants-13-00812-f003:**
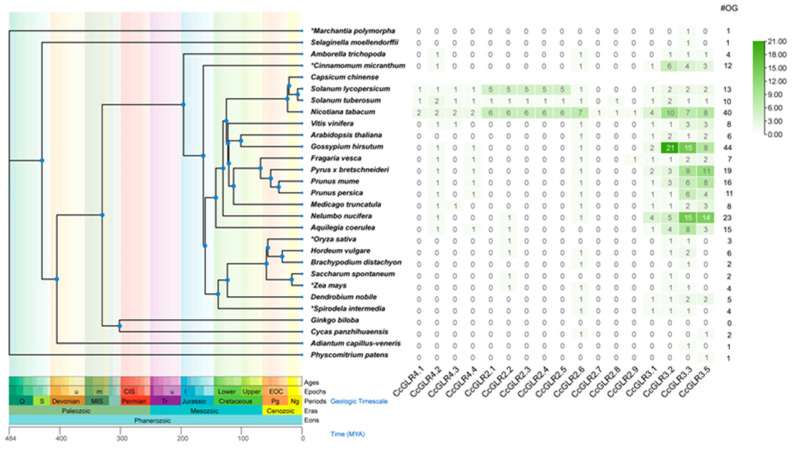
Taxonomic tree of divergent species and identification of orthologous sequences to CcGLRs. MEGA 10.2 was employed for constructing the taxonomic tree. The ortholog analysis was conducted using OrthoVenn with the 17 CcGLR sequences. In total, 408 orthologous protein sequences were identified. * Unresolved names; *M. polymorpha* (replaced with *Marchantia paleacea*), *Cinnamomum micranthum* (replaced with *Persea americana*), *O. sativa* (replaced with *Oryza longistaminata*), *Z. mays* (replaced with *Zea diploperennis*), *Spirodela intermedia* (replaced with *Lemna gibba*).

**Figure 4 plants-13-00812-f004:**
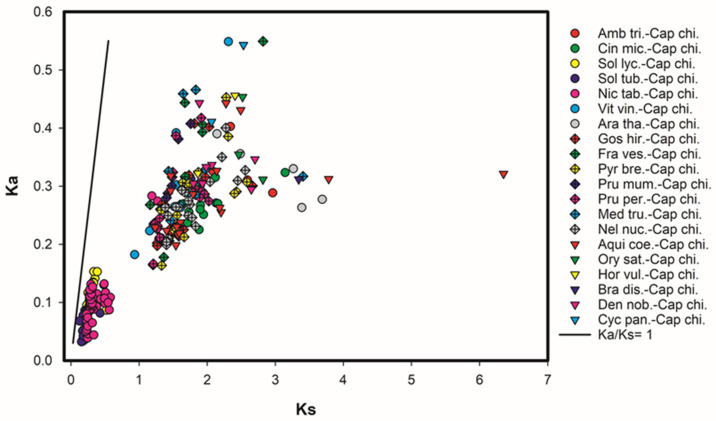
Analysis of Ka and Ks across all species. The *Y*-axis represents non-synonymous mutations (Ka), the *X*-axis represents synonymous mutations (Ks), and the line within the graph represents the Ka/Ks value of 1. Species abbreviations: Amb tri: *A. trichopoda*, Cin mic: *C. micranthum*, Sol lyc: *S. lycopersicum*, Sol tub: *S. tuberosum*, Nic tab: *N. tabacum*, Vit vin: *Vitis vinífera*, Ara tha: *A. thaliana*, Gos hir: *G. hirsutum*, Fra ves: *F. vesca*, Pyr bre: *P. bretchneideri*, Pru mum: *P. mume*, Pru per: *P. pérsica*, Med tru: *M. truncatula*, Nel nuc: *N. nucifera*, Aquí coe: *A. coerulea*, Ory Sat: *O sativa*, Hor vul: *Hordeum vulgare*, Bra dis: *Brachypodium distachyon*, Den nob: *Dendrobium nobile*, Cyc pan: *C. panzhihuaensis*, Cap chi: *C. chinense*.

**Figure 5 plants-13-00812-f005:**
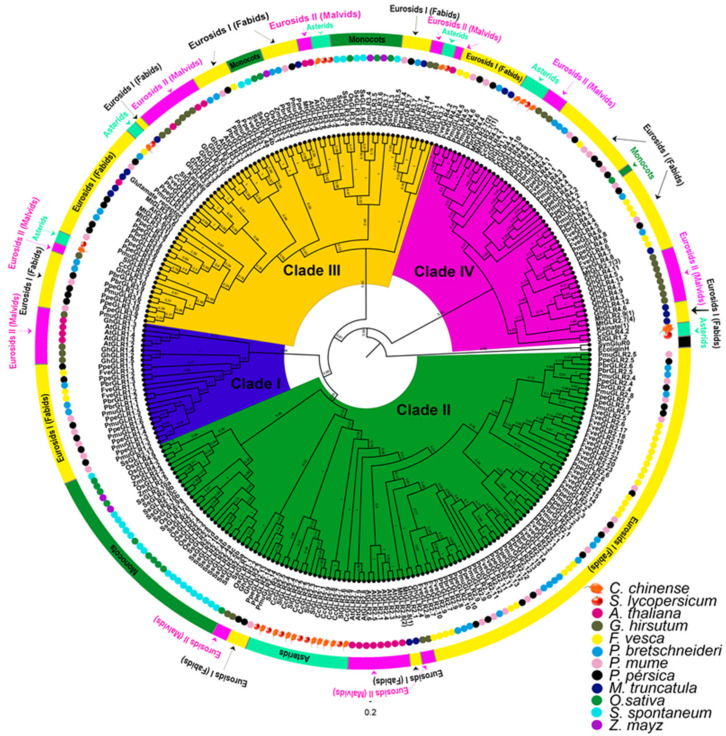
Phylogenetic tree based on amino acid sequence alignments of *C. chinense* GLRs and 11 other previously reported species, totaling 284 sequences: 17 CcGLRs from *C. chinense*, 13 GLRs from *S. lycopersicum* (SlGLR), 20 GLRs from *A. thaliana* (AtGLR), 16 GLRs from *O. sativa* (OsGLR), 15 GLRs from *M. truncatula* (MtGLR), 143 GLRs from woody plants *P. bretschneideri* (33 PbrGLR), *F. vesca* (36 FveGLR), 34 GLRs from *P. mume* (34 PmuGLR), and *P. persica* (40 PpeGLR), 25 GLRs from S. spontaneum (SsGLR), 28 GLRs from *G. hirsutum* (GhGLRs), and 5 GLRs from *Z. mays* (ZmGLR). The analysis was performed using MEGA6. A glutamate receptor from the cyanobacterium *Synechocystis* sp. (SynGluR0) and a bacterial periplasmic amino acid-binding protein GlnH (EcoliglnH) were included as outgroups. The tree was constructed using the Maximum Likelihood method, employing the p-distance model with pairwise deletion, and Bootstrap values were obtained from 1000 repetitions.

**Figure 6 plants-13-00812-f006:**
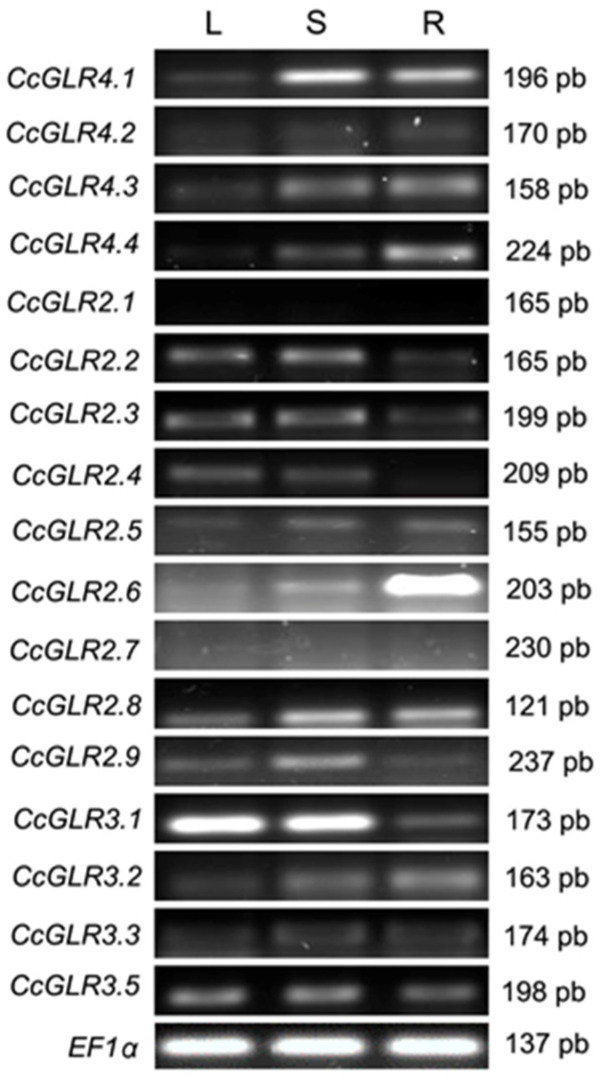
Transcription patterns of all 17 *CcGLRs*, as revealed by endpoint RT-PCR. Samples from leaf (L), stem (S), and root (R) from 4-week-old habanero pepper plants growing in hydroponic conditions were used. The experiment was repeated twice observing consistent results. The EF1a was used as a charge control.

**Figure 7 plants-13-00812-f007:**
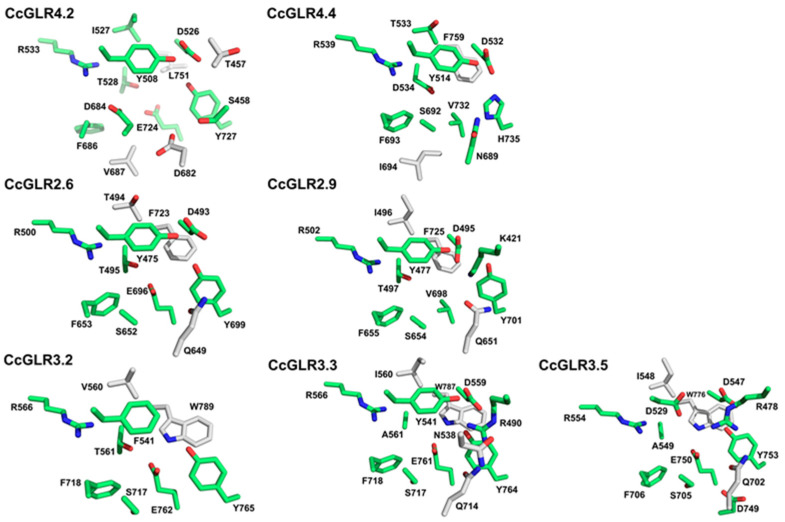
View of the ligand-binding site of 7 CcGLRs. The represented amino acids are those closest to the ligand within a distance of 12 Å. Green-colored side chains represent amino acids previously reported [32], while gray-colored side chains represent those amino acids suggested to be important for ligand binding in habanero pepper GLRs.

**Figure 8 plants-13-00812-f008:**
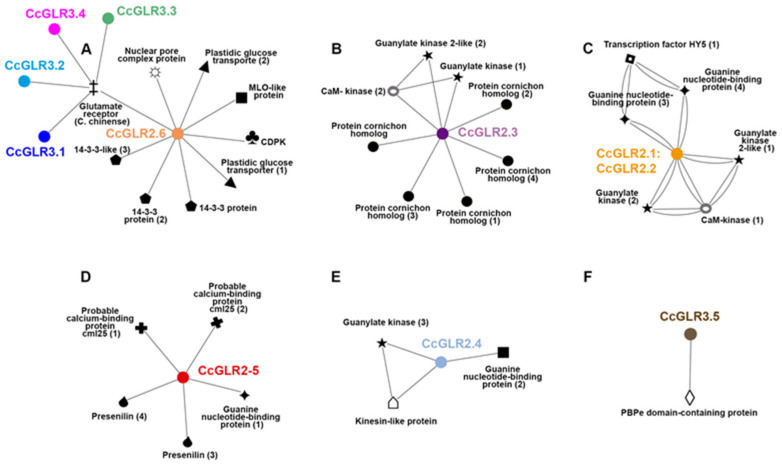
In silico protein–protein interaction. (**A**) Network composed of CcGLR2.6, CcGLR3.1, CcGLR3.2, CcGLR3.3, and CcGLR3.4. (**B**) Network composed of CcGLR2.3. (**C**) Network composed of CcGLR2.1 and CcGLR2.2. (**D**) Network composed of CcGLR2.5. (**E**) Network composed of CcGLR2.4. (**F**) Network composed of CcGLR3.5. Colors represent different proteins with which each reference protein interacts in each CcGLR.

**Table 1 plants-13-00812-t001:** Nomenclature, accession number, and structural characteristics of the GLR family in *Capsicum chinense*.

Protein ID	Protein Name	# Exons	# Introns	CDS (bp)	Amino Acids (aa)	pI	Mw (kDa)	Subcellular Localization	Signal Peptide
PHU20802.1	CcGLR4.1	8	7	2703	901	8.34	100.07	MP	1–25
PHU26278.1	CcGLR4.2	5	4	2784	928	6.72	104.44	MP	1–24
PHU20786.1	CcGLR4.3	5	4	2703	901	8.23	99.93	MP	1–21
PHU20798.1	CcGLR4.4	5	4	2703	901	8.69	100.18	MP	1–22
PHU14992.1	CcGLR2.1	5	4	2823	941	5.95	104.60	MP	NOT
PHU14985.1	CcGLR2.2	5	4	2898	966	5.88	107.48	MP	NOT
PHU14982.1	CcGLR2.3	5	4	2934	978	6.72	109.34	MP	NOT
PHU14980.1	CcGLR2.4	5	4	2874	958	6.48	106.73	MP	NOT
PHU14979.1	CcGLR2.5	6	5	3396	1132	6.2	125.73	MP/Nu’	NOT
PHU08778.1	CcGLR2.6	6	5	2739	913	6.12	101.69	MP	NOT
PHU24790.1	CcGLR2.7	9	8	3336	1112	8.63	124.04	MP	1–15
PHU24792.1	CcGLR2.8	5	4	2913	971	8.67	108.06	MP	1–21
PHU24793.1	CcGLR2.9	5	4	2745	915	6.76	101.43	MP	NOT
PHU12201.1	CcGLR3.1	6	5	2721	907	6.94	101.78	MP	1–19
PHU12202.1	CcGLR3.2	6	5	2829	943	6.22	104.76	MP	1–27
PHU21095.1	CcGLR3.3	6	5	2835	945	8.26	104.86	MP	1–33
PHU26821.1	CcGLR3.5	6	5	2808	936	6.51	104.26	MP	1–18

#: number.

**Table 2 plants-13-00812-t002:** Duplicated CcGLRs in the habanero pepper genome and analysis of their Ka/Ks ratio.

Genes	Genes	Ka	Ks	Ka_Ks	Type of Duplication
CcGLR4.4	CcGLR4.3	0.092388823	0.215387561	0.428942242	Tandem
CcGLR2.2	CcGLR2.1	0.09404683	0.292663192	0.321348336	Tandem
CcGLR2.2	CcGLR2.3	0.081302231	0.281706402	0.288606258	Tandem
CcGLR2.3	CcGLR2.7	0.486844643	2.313179137	0.210465603	Segmental
CcGLR2.3	CcGLR2.4	0.117837234	0.321347488	0.36669723	Tandem
CcGLR2.4	CcGLR2.5	0.136764142	0.328760158	0.415999746	Tandem
CcGLR2.7	CcGLR2.8	0.038893102	0.074861737	0.519532455	Tandem
CcGLR2.8	CcGLR2.9	0.275425921	1.216754722	0.226361087	Tandem
CcGLR3.1	CcGLR3.2	0.469517865	2.611359743	0.179798232	Tandem
CcGLR3.3	CcGLR3.5	0.315604474	1.822451838	0.173175756	Segmental

**Table 3 plants-13-00812-t003:** Orthologs in habanero pepper based on the proteins that interact with CcGLRs previously observed in STRING.

String Protein	Amino Acids (AA)	Description	Orthologs in *C. chinense*	Amino Acids (AA)	Description
LOC104246719 GB1 Solyc01g109560.2.1 LOC104228583	377	Guanine nucleotide-binding protein subunit beta-1	PHU10599.1	377	Guanine nucleotide-binding protein subunit beta
LOC104224515 XP_009606158.1 Solyc02g068340.2.1	1265	Kinesin-like calmodulin-binding protein	PHU25623.1	1505	Kinesin-like calmodulin-binding protein
LOC104214115 XP_009623672.1 Solyc11g005900.1.1	301	Guanylate kinase 2	PHU00386.1	301	Guanylate kinase 3
XP_009589304.1	139	Protein cornichon homolog 4-like	PHU14641.1	139	Protein cornichon -like protein 4
XP_009607364.1	134	Protein cornichon homolog 4-like	PHU17864.1	134	Protein cornichon -like protein 4
XP_009604924.1	135	Protein cornichon homolog 4-like	PHU29157.1	134	Protein cornichon -like protein 4
XP_009616174.1 LOC104216467	402	Guanylate kinase	PHU22297.1	424	Guanylate kinase 1
PGSC0003DMT400063334	431	Presenilin	PHU23970.1	435	Presenilin-like protein
PGSC0003DMT400075801	194	Calcium-binding pollen allergen	PHU17438.1	191	Putative calcium-binding protein CML15
PGSC0003DMT400072494	426	Presenilin	PHU05860.1	435	Presenilin-like protein
A0A2G3D340	116	14-3-3-like protein B	PHU25375.1	116	14-3-3-like protein B
A0A2G3D887	310	Plastidic glucose transporter 4	PHU27196.1	310	Plastidic glucose transporter 4
A0A2G3D8D4	230	Plastidic glucose transporter 4	PHU27195.1	230	Plastidic glucose transporter 4
A0A2G3CNZ2	256	14-3-3 protein 10	PHU20451.1	256	14-3-3 protein 10
A0A2G3CE77	564	Calcium-dependent protein kinase 1	PHU17048.1	564	Calcium-dependent protein kinase 1
A0A2G3BWA2 A0A2G2XW54	604	Glutamate receptor 3.2	PHU10769.1	604	Glutamate receptor 3.2
A0A2G3BX92	446	Nuclear pore complex protein NUP43	PHU11111.1	446	Nuclear pore complex protein NUP43
A0A2G3B0C0	251	14-3-3 protein 10	PHT99929.1	251	14-3-3 protein 10
A0A2G3BE56	425	MLO-like protein	PHU04736.1	425	MLO-like protein 4
LOC104219342	162	Transcription factor HY5 isoform X1	PHU22755.1	158	Transcription factor HY5

## Data Availability

The data are included in the manuscript.

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
