# Peer review of "In Silico Analysis of Glutamate Receptors in Capsicum chinense: Structure, Evolution, and Molecular Interactions"

_plants, 2024, doi:10.3390/plants13060812_

Round 1

Reviewer 1 Report

Comments and Suggestions for Authors
  1. In the abstract, line no.3 “Knowledge of these proteins”, which protein? Since only one glutamate receptor has been mentioned. Please clarify this sentence/line.
  2. Typo error, in the introduction, Arabidopsis genes into three different clades (clades I, II, and II), it should be III.
  3. I suggest a major rewrite of the introduction. Since it does not contain any information about Capsicum Chinese. It should provide an overview of Capsicum Chinese.
  4. The manuscript contains several uniformity errors. For instance, in line 120 (Conserved Domain Search platform) either write all words in small or all in capital letters. Furthermore, in lines 168-169, please use the percentage marks in the entire manuscript to maintain uniformity in the manuscript.
  5. Authors should maintain the citation uniformity. More than one reference could be cited in a single bracket. For instance, line 76, 86.
  6. The authors are advised to maintain the uniformity in entire manuscript while writing Capsicum Chinese. I would suggest the authors to either Capsicum Chinese or habanero pepper. For instance, in results line 111, materials and methods line 612.
  7. I suggest correcting the bibliography uniformity. For instance, reference 44, 46, 66.
  8. Please clarify the sentence “Of the 17 CcGLRs sequences 10 that are grouped in tandem on three different chromosomes (3, 4 and 6), this is 58% of the sequences”. 
  9. The conclusion part is missing in the manuscript. Authors are suggested to conclude their study by writing a separate Section conclusion.
  10. Rational behind selection of Capsicum Chinese and Plant glutamate receptors should be mentioned in the introduction section.
  11. Limitations of current study should be highlighted.

Author Response

Mérida, Yucatán, 27th February 2024

Dr. Bertrand Hirel

Section Editor-in-Chief

Plant Physiology and Metabolism

Dear Dr. Hirel:

Please find enclosed the version of revised manuscript entitled In Silico Analysis of Glutamate Receptors in Capsicum chinense: Structure, Evolution and Molecular Interactions” of Fabiola León-García, Federico García-Laynes, Georgina Estrada-Tapia, Miriam Monforte-González, Manuel Martínez-Estevez, and Ileana Echevarría-Machado *

The manuscript was reviewed by two colleagues and their comments were considered in this latest version. We think that this new version of the manuscript is ready to be published.

In this work we expand the knowledge of the plant glutamate receptor protein family, GLR, using Capsicum chinense Jacq as a model (habanero pepper), a Solanaceae highly demanded in the market. The peculiarities of some CcCGLR in terms of structural characteristics of their domains, orthology, phylogeny, as well as molecular coupling make these proteins unique to suggest them as excellent candidates for functionality studies. We appreciate that you consider the manuscript to be published in this journal.

Below, we detail the responses to the reviewers' comments.

Reviewer 1

Comments and Suggestions for Authors

  1. In the abstract, line no.3 “Knowledge of these proteins”, which protein? Since only one glutamate receptor has been mentioned. Please clarify this sentence/line.

  1. 1.1/ This mistake was corrected in the manuscript. See lines 11-13 .

  1. Typo error, in the introduction, Arabidopsis genes into three different clades (clades I, II, and II), it should be III.

  1. 1.2/ This mistake was corrected in the manuscript. See line 53.

  1. I suggest a major rewrite of the introduction. Since it does not contain any information about Capsicum Chinese. It should provide an overview of Capsicum Chinese.

  1. 1.3/ As the reviewer suggested, the introduction was rewritten and now an overview of Capicum chinense was added. See lines 98-123.

  1. The manuscript contains several uniformity errors. For instance, in line 120 (Conserved Domain Search platform) either write all words in small or all in capital letters. Furthermore, in lines 168-169, please use the percentage marks in the entire manuscript to maintain uniformity in the manuscript.

  1. 1.4/ The manuscript was carefully revised, and these errors were corrected in the text. See lines 149-150, 194, and 198-199.

  1. Authors should maintain the citation uniformity. More than one reference could be cited in a single bracket. For instance, line 76, 86.

  1. 1.5/ The mistake in citation was corrected in the manuscript, as a reviewer suggested. See line 76 and 87.

  1. The authors are advised to maintain the uniformity in entire manuscript while writing Capsicum Chinese. I would suggest the authors to either Capsicum Chinese or habanero pepper. For instance, in results line 111, materials and methods line 612.

  1. 1.6/ The use of Capsicum chinense or habanero pepper was corrected throughout the text, using the scientific name in the headings and the common name in the text. For example, see 139-149.

  1. I suggest correcting the bibliography uniformity. For instance, reference 44, 46, 66.

  1. 1.7/ The bibliography was corrected in the manuscript. See lines 998-1260.

  1. Please clarify the sentence “Of the 17 CcGLRs sequences 10 that are grouped in tandem on three different chromosomes (3, 4 and 6), this is 58% of the sequences”. 

  1. 1.8/ The redaction of this sentence was corrected in the manuscript. See lines 593-594.

  1. The conclusion part is missing in the manuscript. Authors are suggested to conclude their study by writing a separate Section conclusion.

  1. 1.9/ The conclusions section was added in the manuscript, as a reviewer suggested. See lines 956-972.

  1. Rational behind selection of Capsicum Chinese and Plant glutamate receptors should be mentioned in the introduction section.

  1. 1.10/ We appreciate this comment from the reviewer and their suggestion was addressed in the introduction section. See lines 121-136.

  1. Limitations of current study should be highlighted.

  1. 1.11/ The Limitations of current study were added in the manuscript. See lines 509-512, 563-574, 667-683, 715-725, 798-803, and 967-971.

Reviewer 2 Report

Comments and Suggestions for Authors

The manuscript offers a comprehensive analysis of the GLR gene family in the Capsicum chinense genome, covering various aspects such as gene identification, structural characteristics, evolutionary history, protein interactions, and ligand binding properties. However, some major points could enhance the completeness and robustness of the study:

1.     The manuscript would benefit from experimental validations to confirm the functional roles of the identified CcGLRs. This could include expression studies under different conditions, tissue-specific expression patterns or functional assays to understand their roles in signaling pathways.

2.     While the subcellular localization of CcGLRs is discussed, an analysis of their expression profiles under various stimuli or stress conditions would provide insights into their dynamic roles. Understanding how these genes respond to environmental cues or stressors is crucial for elucidating their functional significance.

3.     Discussion needs to be elaborated. Linking the findings to the biological significance of the GLR gene family in Capsicum chinense is essential. Any observed phenotypic traits associated with these genes, such as resistance to specific stresses or developmental characteristics, should be discussed to provide a more comprehensive understanding.

4.     The manuscript lacks a comparative analysis with GLR gene families in other plant species. A discussion on similarities or differences in the structure, function, and evolution of GLRs between Capsicum chinense and other plants within the Solanaceae family or beyond would strengthen the study's context.

5.     While molecular docking analysis is performed, a more detailed discussion on the specific amino acid residues involved in ligand interactions, along with their functional significance, would provide a deeper understanding of the ligand-binding properties of CcGLRs.

6.     While protein-protein interaction networks are predicted, experimental validations through techniques like co-immunoprecipitation or yeast two-hybrid assays would strengthen the reliability of these interactions.

Author Response

Mérida, Yucatán, 27th February 2024

Dr. Bertrand Hirel

Section Editor-in-Chief

Plant Physiology and Metabolism

Dear Dr. Hirel:

Please find enclosed the version of revised manuscript entitled In Silico Analysis of Glutamate Receptors in Capsicum chinense: Structure, Evolution and Molecular Interactions” of Fabiola León-García, Federico García-Laynes, Georgina Estrada-Tapia, Miriam Monforte-González, Manuel Martínez-Estevez, and Ileana Echevarría-Machado *

The manuscript was reviewed by two colleagues and their comments were considered in this latest version. We think that this new version of the manuscript is ready to be published.

In this work we expand the knowledge of the plant glutamate receptor protein family, GLR, using Capsicum chinense Jacq as a model (habanero pepper), a Solanaceae highly demanded in the market. The peculiarities of some CcCGLR in terms of structural characteristics of their domains, orthology, phylogeny, as well as molecular coupling make these proteins unique to suggest them as excellent candidates for functionality studies. We appreciate that you consider the manuscript to be published in this journal.

Below, we detail the responses to the reviewers' comments.

Reviewer 2

Comments and Suggestions for Authors

The manuscript offers a comprehensive analysis of the GLR gene family in the Capsicum chinense genome, covering various aspects such as gene identification, structural characteristics, evolutionary history, protein interactions, and ligand binding properties. However, some major points could enhance the completeness and robustness of the study:

  1. The manuscript would benefit from experimental validations to confirm the functional roles of the identified CcGLRs. This could include expression studies under different conditions, tissue-specific expression patterns or functional assays to understand their roles in signaling pathways.
  2. 2.1/ Although the objective of this study was not to evaluate the transcriptional profiles of the GLRs of habanero pepper, we attended this suggestion of the reviewer and the transcription patterns of all CcGLR genes from leaf, stem and root tissues were added at manuscript. See lines 340-359, 647-683, and 870-883.
  3. While the subcellular localization of CcGLRs is discussed, an analysis of their expression profiles under various stimuli or stress conditions would provide insights into their dynamic roles. Understanding how these genes respond to environmental cues or stressors is crucial for elucidating their functional significance.
  4. 2.2/ We know that the subcellular localization of CcGLRs can contribute to understanding the function of these proteins in the plant. Unfortunately, very few GLR proteins have been localized in the cell, even for those of the model plant Arabidopsis. The objective of this work was to provide a first approach to understanding these proteins in a species that has not yet been studied. In subsequent studies this topic should be addressed, as well as an approach to their function through expression analysis that the reviewer proposes. In this new version, we add the transcript profiles in vegetative organs (see R.2.1), as well as a greater discussion about the localization of these proteins, according to the results obtained. See lines 541-574
  5. Discussion needs to be elaborated. Linking the findings to the biological significance of the GLR gene family in Capsicum chinense is essential. Any observed phenotypic traits associated with these genes, such as resistance to specific stresses or developmental characteristics, should be discussed to provide a more comprehensive understanding.
  6. 2.3/ The discussion was modified, considering the reviewer's comments. In this new version of the manuscript, further discussion about conservation and/or divergence of the most important domains in protein function, as well as their cellular localization, transcription profiles, molecular docking, and protein-protein interaction networks, is contained. See lines 466-803.
  7. The manuscript lacks a comparative analysis with GLR gene families in other plant species. A discussion on similarities or differences in the structure, function, and evolution of GLRs between Capsicum chinense and other plants within the Solanaceae family or beyond would strengthen the study's context.
  8. 2.4/This comparative analysis was added in this manuscript version, as the reviewer suggested. See lines 497-540, 541-574, 647-683, 684-725, and 787-803, in discussion section.
  9. While molecular docking analysis is performed, a more detailed discussion on the specific amino acid residues involved in ligand interactions, along with their functional significance, would provide a deeper understanding of the ligand-binding properties of CcGLRs.
  10. 2.5/ This reviewer's comment was addressed in the manuscript. See lines 684-725.
  11. While protein-protein interaction networks are predicted, experimental validations through techniques like co-immunoprecipitation or yeast two-hybrid assays would strengthen the reliability of these interactions.
  12. 2.6/ We greatly appreciate the reviewer's comment and agree with it; knowing the interaction networks in which these proteins participate greatly contributes to elucidating their function in plants. Unfortunately, not even for all Arabidopsis proteins, reported more than 20 years ago, does this knowledge exist. The results obtained here, based on proteins whose interaction has been demonstrated experimentally and the identification of its orthologs in habanero pepper, contribute to strengthening the possible physiological occurrence of these networks, but they have to be demonstrated experimentally in future works. These comments were added to the text. See lines 787-803.

Round 2

Reviewer 1 Report

Comments and Suggestions for Authors

Authors have addressed all my concerns in the revised version of manuscript.

Reviewer 2 Report

Comments and Suggestions for Authors

I recommend the editor to accept the manuscript in its present form.